# Combined Anticancer Effect of Sulfated Laminaran from the Brown Alga *Alaria angusta* and Polyhydroxysteroid Glycosides from the Starfish *Protoreaster lincki* on 3D Colorectal Carcinoma HCT 116 Cell Line

**DOI:** 10.3390/md19100540

**Published:** 2021-09-25

**Authors:** Olesya S. Malyarenko, Timofey V. Malyarenko, Roza V. Usoltseva, Valerii V. Surits, Alla A. Kicha, Natalia V. Ivanchina, Svetlana P. Ermakova

**Affiliations:** G.B. Elyakov Pacific Institute of Bioorganic Chemistry, Far Eastern Branch of the Russian Academy of Sciences, 159 100-let Vladivostok Ave., 690022 Vladivostok, Russia; malyarenko-tv@mail.ru (T.V.M.); usoltseva-r@yandex.ru (R.V.U.); suritsw@yandex.ru (V.V.S.); kicha@piboc.dvo.ru (A.A.K.); ivanchina@piboc.dvo.ru (N.V.I.); swetlana_e@mail.ru (S.P.E.)

**Keywords:** *Alaria angusta*, sulfated laminaran, *Protoreaster lincki*, polyhydroxysteroid glycosides, colorectal carcinoma, HCT 116 cells, 3D spheroids, AKT, apoptosis induction

## Abstract

Colorectal cancer is one of the most frequent types of malignancy in the world. The search for new approaches of increasing the efficacy of cancer therapy is relevant. This work was aimed to study individual, combined anticancer effects, and molecular mechanism of action of sulfated laminaran AaLs of the brown alga *Alaria angusta* and protolinckiosides A (PL1), B (PL2), and linckoside L1 (L1) of the starfish *Protoreaster* *lincki* using a 3D cell culture model. The 3-(4,5-dimethylthiazol-2-yl)-5-(3-carboxymethoxyphenyl)-2-(4-sulfophenyl)-2H-tetrazolium (MTS), soft agar, 3D spheroids invasion, and Western blotting assays were performed to determine the effect and mechanism of the action of investigated compounds or their combinations on proliferation, colony formation, and the invasion of 3D HCT 116 spheroids. AaLs, PL1, PL2, and L1 individually inhibited viability, colony growth, and the invasion of 3D HCT 116 spheroids in a variable degree with greater activity of linckoside L1. AaLs in combination with L1 exerted synergism of a combined anticancer effect through the inactivation of protein kinase B (AKT) kinase and, consequently, the induction of apoptosis via the regulation of proapoptotic/antiapoptotic proteins balance. The obtained data about the efficacy of the combined anticancer effect of a laminaran derivative of brown algae and polyhydroxysteroid glycosides of starfish open up prospects for the development of new therapeutic approaches for colorectal cancer treatment.

## 1. Introduction

Oncological diseases are one of the main causes of human mortality in the modern world, and the economic and financial burden for cancer research is increasing [1]. Colorectal cancer is the most frequently diagnosed malignant cancer around the world, and one of the most common causes of death among cancer patients [2]. Approximately 20 to 50% of patients already have distant metastases at the time of diagnosis [3]. Unfortunately, many radically operated on patients experience dissemination of the process at different periods after the operation, and the overall 5-year survival rate of patients with metastatic colon cancer does not exceed 7–8% [4]. Despite the noticeable progress in the treatment of colorectal cancer using cytotoxic and targeted drugs, the effectiveness of chemotherapy for this type of cancer cannot be considered satisfactory, which dictates the need to search for new approaches in the treatment of patients with advanced colorectal cancer [5]. For instance, the combination of two or more therapeutic treatments specifically regulating cancer-inducing or cell-sustaining pathways can be a prospective approach for improving the scheme of colorectal cancer therapy [6,7,8].

Marine natural compounds such as polysaccharides from brown algae and steroidal compounds from starfish are of constant interest due to their extremely broad spectrum of biological action. Neutral water-soluble 1,3;1,6-β-D-glucans of brown algae (laminarans) and their chemical derivatives possess immunomodulatory, antiproliferative, and radioprotective activities [9]. To date, a few publications are known concerning the antiproliferative, proapoptotic, and radiosensitizing activities of laminaran from brown algae in cancer cells [10,11,12,13,14,15,16].

Natural steroidal compounds represent a large class of secondary metabolites found in various terrestrial and marine organisms, in particular, in marine invertebrates. Mainly, the interest in these compounds is explained by their various biological activities, including anticancer, antiviral, anti-inflammatory, analgesic, hemolytic, hypotensive, neuroprotective, and immunomodulatory properties [17,18,19,20]. Previously, we investigated the immunomodulatory activity of polyhydroxysteroid glycosides: protolinckiosides A (PL1) and B (PL2) and linckoside L1 (L1), including lysosomal stimulation and intracellular reactive oxygen species (ROS) levels regulation on the RAW 264.7 murine macrophage cells. Polyhydroxysteroid glycosides PL1, PL2, and L1 were shown not to exhibit an ability to stimulate lysosomal activity of the RAW 264.7 cells and only PL2 induced ROS formation in the RAW 264.7 cells. Additionally, polyhydroxysteroid glycosides PL1 and PL2, but not L1, reduced the ROS formation in the RAW 264.7 cells induced by the proinflammatory endotoxin LPS from *Escherichia coli* [21].

We hypothesized that the combination of the unique biological properties of the metabolites of brown algae and starfish will contribute to the development of highly effective approaches for the treatment of advanced colorectal cancer.

The aim of the present work was to investigate the individual and combined anticancer effects and molecular mechanism of the action of sulfated laminaran from the brown alga *Alaria angusta* and polyhydroxysteroid glycosides from the starfish *Protoreaster lincki* with focus on the proliferation, colony growth, and invasion of 3D spheroids of colorectal carcinoma HCT 116 cells.

## 2. Results and Discussion

### 2.1. Isolation, Modification of Laminaran from A. angusta, and Structural Characteristics of Sulfated Laminaran

Laminaran from the brown alga *A. angusta* was isolated according to a modified method described in recent work [22]. To remove low molecular weight compounds, the alga was treated with organic solvents and air-dried. Then the extraction was performed with 0.1 N hydrochloric acid at 60 °C for 2 h, dialyzed, concentrated, precipitated with four volumes of 96% ethanol, and air-dried in order to obtain crude fractions of water-soluble polysaccharides (laminarans and fucoidans). A highly purified preparation of laminaran was obtained during anionexchange chromatography on a Macro-prep DEAE and hydrophobic chromatography on polychrome. The structural characteristics of native laminaran AaL from *A. angusta* were determined using high-performance liquid chromatography (HPLC), chemical (methods for determining the content of total carbohydrates and sulfates), and ^13^C NMR spectroscopy (Table 1, Appendix A). It was confirmed that laminaran AaL has a backbone of 1,3-linked β-D-glucopyranose residues with single glucose branches at C-6 with a bond ratio of 1,3:1,6 = 6:1 (Appendix A). The laminarans with similar structural characteristics were isolated from the brown algae *Dictyota dichotoma* (order Dictyotales) [23], *Alaria marginata* [22], *Saccharina cichorioides* and from *Saccharina japonica* [14], *Saccharina gurjanovae* [24] (order Laminariales), *Coccophora langsdorfii* [25], *Sargassum duplicatum* [26], and *Sargassum fusiforme* [27] (order Fucales).

To enhance the anticancer activity of the native laminaran from *A. angusta*, it was subjected to a chemical modification. The sulfated laminaran was obtained as a result of the reaction of nucleophilic substitution of hydroxyl groups of the molecules of laminaran from *A. angusta* by sulfate groups using a chlorosulfonic-pyridine complex as a sulfating agent. The yield of obtained sulfated laminaran AaLs was 89%. The structural characteristics of sulfated laminaran AaLs are presented in Table 1. The content of the sulfate groups of laminaran’s derivative was determined by the turbidimetrical method. The sulfated laminaran AaLs from the brown alga *A. angusta* was sulfated by 44% (Table 1).

The structure of sulfated laminaran AaLs was investigated by ^13^C NMR spectroscopy. The downfield location of signals C-2, C-4, and C-6 at 80.5–81.5, 75.9, and 69.3 ppm corresponded that modified polysaccharide has sulfate groups at position 2, 4, and 6, respectively (Appendix A).

### 2.2. The Polyhydroxysteroid Glycosides from the Starfish P. lincki

Polyhydroxysteroid glycosides PL1, PL2, and L1 were previously isolated from the starfish *P. lincki* according to the scheme described earlier [21].

It was identified that all of the isolated compounds are steroidal monoglycosides, but differ from each other by the steroid nucleus and monosaccharide residues. Thus, protolinckioside A (PL1) has a 3β,4β,6β,7α,8,15α,16β,26-octahydroxysteroid nucleus, protolinckioside B (PL2) has a 3β,5,6β,8,15α,24-hexahydroxysteroid nucleus, and linckoside L1 (L1) has a 3β,4β,6β,8,15α,24-hexahydroxysteroid nucleus. The monosaccharide residues in PL1 and L1 are the 2-O-methyl-β-D-xylopyranose attached to C-3 of the steroid nucleus, and in PL2—the α-L-arabinofuranose attached to C-24 of the side chain of the steroid aglycone (Figure 1).

### 2.3. The Individual and Combined Effects of Sulfated Laminaran and Polyxydroxysteroid Glycosides on Cell Viability and Proliferation of 3D HCT 116 Spheroids

Currently, the two-dimensional cell culture (2D culture) is a widely used in vitro model in cell biology. The success of this model is due to the property of most types of animal and human cells to proliferate in special culture media and form homogeneous colonies. However, the cell growth of a monolayer culture in many parameters does not reflect the true picture of tumor growth in a living organism, where interactions not only between the cells of the tumor, but also with the surrounding extracellular matrix, represented in its core by connective tissue cells and collagen cells, are of great importance in its progression. The three-dimensional cell culture, represented by spheroids (3D spheroids), is the most effective system that is as close as possible in properties and organization to a tumor in organisms, which is currently used for screening potential anticancer drugs [28].

The cytostatic activity of sulfated derivative of laminaran AaLs and polyhydroxysteroid glycosides PL1, PL2, and L1 was studied using a 3D cell culture model of human colorectal carcinoma cells HCT 116 (3D HCT 116 spheroids) by 3-(4, 5-dimethylthiazol-2-yl)-5-(3-carboxymethoxyphenyl)-2-(4-sulfophenyl)-2H-tetrazolium (MTS) assay. PL1 at concentrations of 6.25, 12.5, 25, 50, 100, and 200 μM was found to reduce the cell viability by 17%, 19%, 15%, 18%, 31%, and 32%, respectively, but the concentration of compounds causing 50% inhibition of the spheroids’ viability (IC_50_) was not achieved in the concentration range up to 200 μM even in 3 days of treatment (Figure 2A). It was shown that PL2 has a comparable cytostatic effect as PL1 and insignificantly affected the viability of 3D HCT 116 spheroids at concentrations up to 200 μM (Figure 2B). The IC_50_ of linckoside L1 was determined to be 111.4, 100.0, and 89.3 μM after the 3D HCT 116 spheroids’ treatment for 24 h, 48 h, and 72 h, respectively (Figure 2C). It was found previously that compounds PL1 and L1 possessed cytotoxic activity against RAW 264.7 murine macrophage cells [21]. PL1 and L1 at a concentration of 100 µM reduced the cell viability by 20% and 50%, respectively, in 24 h of incubation. At the same time, polyhydroxysteroid glycoside PL2 did not demonstrate significant cytotoxic effects within a concentration range of 0.001–100.0 µM [21].

The sulfated laminaran’s derivative AaLs exhibited a moderate antiproliferative effect against 3D HCT 116 spheroids growth at a concentration range from 400 to 1600 μg/mL with IC_50_ value of 1555.6, 746.7, and 469.6 μg/mL after the incubation of the spheroids for 24 h, 48 h, and 72 h, respectively (Figure 2D). Several scientific groups have studied the effect of laminaran from the brown alga *Laminaria digitata* (commercially available from Sigma Aldrich (Saint Louis, MO, USA)) on the proliferation of different types of human cancer cells. Thus, it was shown that laminaran (5 mg/mL) from the brown alga *L. digitata* inhibited the proliferation of colorectal cancer cells HT-29 by 40% without affecting the growth of normal intestinal epithelial cells EC-6 [11]. The molecular mechanism of the antiproliferative action of laminaran was found to be associated with the regulation of members of the Fas and IGF-IR signaling pathways and the inducing of intrinsic apoptotic and ErbB pathways involved in carcinogenesis and neurodegenerative disorders [11]. It was also reported that sulfated laminaran, obtained by the chemical modification of laminaran from *L. digitata*, reduced the growth rate of PC-3 human prostate cells, arrested the cell cycle in the S phase, and induced apoptosis in cancer cells through the regulation of the expression level of PTEN and P271kip1 proteins [10]. Ji Y.B. and et al. showed that laminaran was able to induce the apoptosis of the human intestinal cancer cells Lovo by regulating the expression level of death receptors DR4, DR5, TRAIL, FADD, apoptotic proteins Bid, tBid, Bax, Bcl-2, and caspases 8 and 3 [12]. Additionally, the sulfated derivative of laminaran from *L. digitata* was obtained and its antiproliferative activity was determined. The sulfated laminaran (at a dose of 1.6 mg/mL) was found to have a stronger inhibitory effect on the proliferation of Lovo cancer cells than the native polysaccharide [13]. The authors suggested that the introduction of sulfate groups into the motif of laminarans changed their physicochemical properties, three-dimensional conformation and, consequently, enhanced the anticancer activity compared to native laminaran. The antitumor activity of native laminarans isolated from the brown algae *S. cichorioides*, *S. japonica*, and *Fucus evanescens*, as well as their sulfated derivatives, was studied by our scientific group [14]. It was confirmed that sulfated derivatives of laminarans possessed higher anticancer activity in vitro than native polysaccharides. The sulfated derivative of laminaran from *F. evanescens* was shown to have more pronounced anticancer activity among sulfated laminarans from *S. cichorioides* and *S. japonica* and effectively inhibited proliferation, colony formation, and migration of triple-negative breast cancer cell line MDA-MB-231 [14].

In our study, the concentrations of polyhydroxysteroid glycosides PL1, PL2, and L1 and sulfated laminaran AaLs for the combined treatment of 3D HCT 116 spheroids were chosen based on the preliminary experimental data (Appendix A, data is represented for L1 and AaLs). So, all polyhydroxysteroid glycosides were tested in minimally inhibiting concentrations of 6.25 and 12.5 µM at which, individually, they slightly affected the viability of spheroids (the percentage of inhibition of cell viability was less than 20%) (Appendix A). As for sulfated laminaran, based on the previous published data on the effect of the combined effect of sulfated laminaran from *Dictyota dichotoma* with X-ray radiation, where the radiosensitizing activity of sulfated laminaran was observed at low concentrations of 10, 20, and 40 µg/mL [16], we checked the ability of AaLs to potentiate the activity of PL1, PL2, and L1 at concentrations of 10, 20, 40, 80, and 160 µg/mL, which are much lower than its IC_50_ (Appendix A). A strong synergism of anticancer effect was observed after the treatment of 3D HCT 116 spheroids by L1 at 12.5 µM and AaLs at 10, 20, and 40 µg/mL. The significant difference between the efficacy of the combined action of L1 (12.5 μM) and AaLs at higher concentrations of 80 and 160 μg/mL was not found compared to the action of AaLs at low concentrations of 10, 20, and 40 μg/mL. Therefore, we further investigated the combined effect of polyhydroxysteroid glycosides at 12.5 µM and sulfated laminaran at 10, 20, and 40 µg/mL.

It was shown that PL1 or PL2 in combination with AaLs at low sub-cytotoxic concentrations did not influence viability or size of 3D HCT 116 spheroids (Figure 3A,B). On the other hand, the sulfated laminaran AaLs was found to enhance the antiproliferative effect of polyhydroxysteroid glycoside L1 (Figure 3C). The type of interaction of L1 and AaLs was determined by the Chou–Talalay method for drug combination. The combination index (CI) or fraction affected (Fa) indicated the degree of drug interaction was 0.54, 0.34, and 0.33 or 0.68, 0.59, and 0.52, respectively, which confirmed the synergism of the combined anticancer effect of L1 with AaLs on viability of the 3D HCT 116 spheroids (Figure 3D).

### 2.4. The Individual and Combined Effects of Sulfated Laminaran and Polyhydroxysteroid Glycosides on Colony Formation of 3D HCT 116 Spheroids

The individual and combined effects of polyhydroxysteroid glycosides PL1, PL2, L1 and sulfated laminaran AaLs on the model of growth of the colony of 3D HCT 116 spheroids were determined in soft agar. It was shown that PL1 at 12.5, 25, and 50 μM by itself inhibited growth of the colony by 4%, 9%, and 24%, respectively, compared to the nontreated spheroids (control) (Figure 4A). The individual inhibiting effect of steroidal monoglycoside PL2 on the colony growth of 3D HCT 116 spheroids was weak at concentrations up to 50 μM; the percentage of inhibition was less than 15% compared to the control (Figure 4A). L1 at the same doses was found to effectively reduce the size of 3D HCT 116 spheroids by 16%, 38%, and 61%, respectively, compared to the control (Figure 4A). Sulfated laminaran AaLs at concentrations of 100, 200, and 400 μg/mL was demonstrated to inhibit the growth of the colony of 3D cell cultures by 6%, 20%, and 30%, respectively (Figure 4B).

PL1 (12.5 μM) in combination with AaLs (10, 20, and 40 μg/mL) inhibited the growth of the colony of 3D HCT 116 spheroids by 16%, 22%, and 23%, respectively, compared with the individual action of PL1 (Figure 4C,D). The combined treatment of spheroids by PL2 with AaLs was insignificant in the model of soft agar (Figure 4E,F). The efficacy of anticancer treatment of 3D HCT 116 spheroids was revealed only for the combination of polyhydroxysteroid glycoside L1 with sulfated laminaran AaLs. It was shown that L1 (12.5 μM) in combination with AaLs at concentrations of 10, 20, and 40 μg/mL possessed a potent anticancer effect, inhibiting the colony growth of 3D spheroids by 25%, 36%, and 80%, respectively, compared with the individual action of L1 (Figure 4G,H). Taking into account the values of the CI (0.46, 0.37, 0.05) and Fa (0.49, 0.42, and 0.04), the synergism of the combined effect of L1 and AaLs was revealed in the model of colony growth in soft agar (Figure 4J).

### 2.5. The Individual and Combined Effects of Sulfated Laminaran and Polyhydroxysteroid Glycosides on Invasion of 3D HCT 116 Spheroids

There are various invasive assays in the literature, most of which differ in the composition of the polymer matrix that forms an artificial barrier. In this work, a commercial preparation of ECM gel (“Sigma-Aldrich”, Saint Louis, MO, USA) was used to create a polymer matrix, which is a standardized mixture of the extracellular matrix proteins obtained from Engelbreth-Holm-Swarm murine sarcoma.

The 3D HCT 116 spheroids were shown to significantly invade into the polymer matrix after 72 h of their incubation (Figure 5A,F). PL1 by itself at concentrations of 12.5, 25.0, and 50.0 μM decreased the invasion area of spheroids by 20%, 23%, and 37%, respectively, compared to the control (Figure 5B,F). PL2 did not influence the process of invasion of 3D HCT 116 spheroids up to 50 μM (Figure 6C,F). The polyhydroxysteroid glycoside L1 at 50 μM possessed the most significant anti-invasive activity and at 50 μM almost completely prevented the invasion of spheroids into the matrix after 72 h of incubation (Figure 6D,F). It was found that sulfated laminaran AaLs at concentrations of 100, 200, and 400 μg/mL individually inhibited the process of the 3D HCT 116 spheroids’ invasion of by 40%, 54%, and 59%, respectively (Figure 5E,F).

Since linckoside L1 demonstrated the highest individual anti-invasive activity among the investigated polyhydroxysteroid glycosides, it was chosen for the study of the combined effect with a sulfated derivative of laminaran AaLs. It was found that the treatment of 3D HCT 116 spheroids by L1 (12.5 μM) in combination with AaLs (10, 20, and 40 μg/mL) led to the reduction of the invasion area of the spheroids by 16%, 26%, and 44%, respectively, compared to the area of spheroids treated by L1 individually (Figure 6A,B). It was shown that derivative of laminaran AaLs at a low concentration of 10 and 20 μg/mL did not potentiate the effect of L1; the type of action was near additive (Figure 6C). The synergistic anti-invasive effect was observed only when 3D HCT 116 spheroids were treated by L1 at 12.5 μM and AaLs at 40 μg/mL (Figure 6C).

Metastasis is a consequence of tumor progression based on the disturbance of survival/apoptosis balance in cancer cells [29]. One of the possible ways of human colorectal carcinoma progression and, consequently, its metastasis is the overexpression and activation of the survival factor serine/threonine kinase, AKT, also known as protein kinase B (PKB). The activation of AKT leads the apoptosis suppression and the uncontrolled proliferation of cancer cells [30,31,32]. AKT is activated by phosphorylation on Thr308 or Ser473 by different stimuli (growth factors, carcinogens, and radiation) and inactivates members of the cell death machinery, such as proteins of the Bcl-2 family [33]. Several members of the Bcl-2 family (including Bcl-2, Bcl-XL, Mcl-1, A1, and Bag-1) promote survival while other members (including Bcl-XS, Bad, Bax, and Bak) promote cell death [34]. It has been proposed that activated AKT phosphorylates Bad at Ser136 block Bad from binding to Bcl-XL or Bcl-2 and promote survival by preventing the concomitant generation of Bax homodimers [35]. So, we supposed that the molecular mechanism of the combined anticancer effect of sulfated derivative of laminaran AaLs from *A. angusta* and linckoside L1 from *P. lincki* is associated with the regulation of the activity of AKT kinase, and the major players of apoptosis, namely proapoptotic (Bax), antiapoptotic (Bcl-XL) proteins, and the executioner of apoptosis—caspase 3 and cleaved caspase 3. As shown in the Figure 6D,E, there was a strong activation of AKT kinase in nontreated 3D HCT 116 spheroids (control). Investigated compounds L1 and AaLs alone did not significantly influence the activity of p-AKT kinase (Ser 473) and apoptotic proteins (Figure 6D,E). On the other hand, L1 (12.5 µM) in combination with AaLs (20 and 40 µg/mL) effectively inhibited phosphorylation of AKT kinase, followed by the up-regulation of the expression level of the proapoptotic protein Bax and the down-regulation of the expression of the antiapoptotic Bcl-XL protein (Figure 6D,E). This leads to the induction of apoptosis by the activation of an effector caspase 3, and cleaved caspase 3 responsible for the cleavage of the number of death substrates leads to the well-known characteristic hallmarks of an apoptotic cell, including DNA fragmentation, nuclear fragmentation, membrane blebbing, and other morphological and biochemical changes. However, further additional experiments, such as the use of PI3K inhibitor, cell cycle analysis, and DNA comet assay, are needed to confirm the involvement of the PI3K/AKT pathway and the induction of apoptosis in the combined anticancer effect of sulfated laminaran from the brown alga *A. angusta* and polyhydroxysteroid glycosides from the starfish *P. lincki*.

In conclusion, we confirmed that the sulfation of laminaran allows increasing its anticancer activity in vitro compared to the activity of native laminaran. Additionally, it was determined that the in vitro anticancer activity of polyhydroxysteroid glycosides PL1 and L1 was more pronounced than the PL2 one in the model of 3D HCT 116 spheroids. PL1 and L1 have similar structural features: 3β,4β-hydroxyl groups in ring A of the steroidal nucleus and the 2-O-methyl-β-D-xylopyranose residue attached to C-3 of the steroid nucleus, while compound PL2 has another ring A of the steroidal nucleus and the α-L-arabinofuranose residue attached to C-24 of the side chain of the steroid aglycone. Probably, indicated above, the structural features of PL1 and L1 contributed to their significant anticancer activity.

The combination of sulfated derivative of laminaran AaLs from the brown alga *A. angusta* with linckoside L1 from the starfish *P. lincki* inhibits colorectal cancer progression in vitro, and it is much more effective than the individual effect of these compounds at low, nontoxic concentrations. Moreover, this study, for the first time, provides strong evidence that AaLs in combination with L1 suppressed the proliferation, colony formation, and invasion of 3D HCT 116 spheroids via blocking Akt kinase activity and triggered the apoptosis of colorectal carcinoma cells.

It enriches our understanding of the molecular mechanism of the anticancer action of polysaccharide derivatives of brown algae and polyhydroxysteroid glycosides of starfish, opening up prospects for the development of new approaches for therapy of advanced colorectal cancer in the future. We hope that the obtained data will be of great interest, particularly to researchers working on marine natural compounds and oncology.

## 3. Materials and Methods

### 3.1. Laminaran and Its Sulfated Derivative

#### 3.1.1. Brown Alga

The sample of the alga *A. angusta* (Aa) was collected from Simushir Island in August 2016, Sea of Okhotsk during a cruise of the research vessel “Academik Oparin”. A thallus of fresh algal biomass (100 g) was washed thoroughly with tap water, powdered, and pretreated with 70% aqueous ethanol (*w*/*v* = 1:10) for 10 days. Then defatted alga was air-dried at room temperature.

#### 3.1.2. Isolation of Laminaran from *A. angusta*

The laminaran AaL was isolated from the brown alga *A. angusta* by the methods as described previously [23].

#### 3.1.3. The Sulfation of Laminaran from *A. angusta*

Native laminaran AaL from *A. angusta* was modified by chlorosulfonic acid/pyridine method to obtain the sulfated laminaran AaLs as described earlier [14].

#### 3.1.4. The Structural Characteristics of Sulfated Laminaran

The content of carbohydrates was determined according to the method of Michel Dubois et al. [36]. The absorbance was measured at 490 nm using a Power Wave XS microplate reader (BioTek, Winooski, VT, USA). The glucose (1 mg/mL) was used as a reference standard.

The content of the sulfate group was determined by the turbidimetric method after hydrolysis of the sulfated laminaran AaLs with HCl (1N) [37]. The absorbance was measured at 360 nm using a Power Wave XS microplate reader (BioTek, Winooski, VT, USA). K_2_SO_4_ (1 mg/mL) was used as a reference standard.

The molecular weight of sulfated laminaran AaLs was determined by size-exclusion chromatography (SEC), using an Agilent 1100 Series HPLC instrument (“Agilent Technologies”, Waldbronn, Germany) equipped with a refractive index detector and series-connected SEC columns, Shodex OHpak SB-805 HQ and OHpak SB-803 HQ, (“Showa Denko”, Tokyo, Japan). Elution was performed with 0.15 M NaCl aqueous solution at 40°C, with a flow rate of 0.4 mL/min. The dextrans of 5, 10, 50, 80, 250, 410, and 670 kDa (“Sigma-Aldrich”, Saint Louis, MO, USA) were used as reference standards.

The ^13^C NMR spectra of native and sulfated laminarans were obtained on an Avance DPX-500 NMR spectrometer (“Bruker BioSpin Corporation”, Billerica, MA, USA) at 50 °C. The sample concentration was 15 mg of polysaccharide/mL of D_2_O.

### 3.2. Polyhydroxysteroid Glycosides from the Starfish P. lincki

#### 3.2.1. Starfish

Specimens of *P. lincki* BLAINVILLE, 1830 (order Valvatida, family Oreasteridae) were collected at a depth of 1–5 m in the Arabian Sea near the city of Thiruvananthapuram in India. The species identification was carried out by Dr. Padmakumar K.P. (Kerala University of Fisheries and Ocean Studies, Kochi, Kerala, India). A voucher specimen (No. 2010-2) was deposited at the G.B. Elyakov Pacific Institute of Bioorganic Chemistry FEB RAS, Vladivostok, Russia.

#### 3.2.2. Polyhydroxysteroid Glycosides Isolation

Polyhydroxysteroid glycosides PL1, PL2, and L1 were isolated according to the scheme described earlier [21]. In brief, the starfish (500 g) were crushed and extracted twice with methanol (1.0 L) and then twice with ethanol (1.0 L) at room temperature. The extract was concentrated under a vacuum and dissolved in water (2.0 L). Then, the compounds were divided on a polychrome column and chromatographed on a silica gel column in a chloroform–ethanol system (stepwise gradient from 5:1 to 1:5). Nine obtained fractions enriched in polar steroid compounds and their glycosides were subsequently chromatographed on a Florisil column using a chloroform–ethanol system (step gradient from 6:1 to 1:2). The resulting subfractions were further chromatographed using HPLC on reverse phase columns: Diasfer-110-C18 (65% EtOH, 2.5 mL min) and Discovery C18 (65% EtOH, 2.5 mL/min). As a result, the individual compounds were isolated: PL1 (3.5 mg, t_R_ 25.2 min), PL2 (2.5 mg, t_R_ 21.9 min), and L1 (3.0 mg t_R_ 8.9 min). The pbtained PL1, PL2, and L1 were identified based on their thin-layer chromatography (TLC; R_f_, system BuOH:EtOH:H_2_O 4:1:2), HPLC (t_R_, Discovery C18 column, 65% EtOH) nuclear magnetic resonance (NMR), and mass spectrometry (MS) data.

### 3.3. Preparation of Compounds for Investigations of In Vitro Anticancer Activity

The vehicle control is the cells treated with the equivalent volume of phosphate-buffered saline (PBS) or dimethyl sulfoxide (DMSO;final concentration was less than 0.5%) for all of the presented experiments.

Sulfated laminaran AaLs was dissolved in sterile PBS, filtered by 0.22 µm membrane (“Millipore”, Billerica, MA, USA) to prepare stock concentrations of 20 mg/mL.

Polyhydroxysteroid glycosides PL1, PL2, and L1 were dissolved in sterile DMSO to prepare stock concentrations of 40 mM.

Cells were treated with serially diluted AaLs (10, 20, 40 µg/mL or 100, 200, 400, 800, 1600 µg/mL) or polyhydroxysteroid glycosides PL1, PL2, and L1 (6.25, 12.5, 25, 50, 100, and 200 µM) (culture medium used as diluent).

### 3.4. In Vitro Anticancer Activity

#### 3.4.1. Cell Lines and Culture Conditions

The human colorectal carcinoma cells HCT 116 (ATCC^®^ CCL-247™) were obtained from the American Type Culture Collection (Manassas, VA, USA). HCT 116 cells were cultured in in McCoy’s 5A medium supplemented by 10% heat-inactivated fetal bovine serum (FBS) and 1% penicillin–streptomycin solution at 37 °C in a humidified atmosphere containing 5% CO_2_. The passage number was carefully controlled and the mycoplasma contamination was monitored on a regular basis.

#### 3.4.2. Formation of Spheroids (3D Cell Culture) by Liquid Overlay Technique (LOT)

The 3D HCT 116 spheroids were formed by the liquid overlay technique (LOT) method with slight modifications. Briefly, to create nonadherent surfaces for the efficient spheroids’ formation, 50 µL of preheated (60 °C) agarose (1.5%) was overlaid the bottom of 96-well plates and left to solidify for 1 h at room temperature under sterile conditions.

HCT 116 cells (2.0 × 10^3^) were inoculated in an agarose layer and cultured in 200 µL of a complete Dulbecco’s modified eagle medium (DMEM)/10% FBS medium for 96 h at 37 °C in a 5% CO_2_ incubator. An image of each spheroid was made with a ZOE™ Fluorescent Cell Imager (“Bio Rad”, Hercules, CA, USA). ImageJ software bundled with 64-bit Java 1.8.0_112 (“NIH”, Bethesda, MD, USA) was used to measure the spheroid integrity, diameter, and volume.

#### 3.4.3. Determination of Cytostatic Activity by MTS Method

The 3D HCT 116 spheroids were treated by replacing 100 µL of supernatant with a complete medium containing polyhydroxysteroid glycosides PL1, PL2, and L1 at 6.25–200 µM or AaLs at 400–1600 µg/mL for 24, 48, and 72 h. Then, 15 µL of 3-(4,5-dimethylthiazol-2-yl)-5-(3-carboxymethoxyphenyl)-2-(4-sulfophenyl)-2H-tetrazolium (MTS) reagent (“Promega”, Madison, WI, USA) was added to each well with spheroids and incubated for 3 h at 37 °C in a 5% CO_2_ incubator. The absorbance of each well was measured at 490/630 nm using Power Wave XS microplate reader (“BioTek”, Wynusky, VT, USA). The concentration at which a compound exerts half of its maximal inhibitory effect on cell viability (IC_50_) was calculated by the AAT-Bioquest^®^ online calculator [38].

To study the combined effect of PL1, PL2 or L1 with AaLs, the 3D HCT 116 spheroids were treated by compounds in fixed concentration ratios, which were chosen according to their IC_50._ Namely, spheroids were pretreated by PL1, PL2 or L1 at 12.5 µL for 24 h and then AaLs at 10, 20, and 40 µg/mL for an additional 48 h. The metabolic activity of the 3D spheroids after the combined treatment were measured by MTS assay as described above.

#### 3.4.4. Determination of Colony-Inhibiting Activity by Soft Agar Assay

The 3D HCT 116 spheroids were treated by replacing 100 µL of supernatant with a complete medium containing individual compounds: polyhydroxysteroid glycosides PL1, PL2, and L1 at 12.5–50 µM and AaLs at 200–800 µg/mL for 24 h, or their combinations PL1, PL2 or L1 at 12.5 µL for 24 h and then AaLs at 10, 20, and 40 µg/mL for an additional 48 h. The spheroids (*n* = 6) from each treatment were collected and applied onto 0.3% basal medium eagle (BME) agar containing 10% FBS, 2 mM L-glutamine, and 25 µg/mL gentamicin. The cultures were maintained at 37 °C, in a 5% CO_2_ incubator for 21 days, and the cell’s colonies were scored using a microscope Motic AE 20 (XiangAn, Xiamen 361101, China) and the ImageJ software.

#### 3.4.5. Determination of Anti-Invasive Activity by 3D Spheroids Invasion Assay

After the 3D HCT 116 spheroids were formed, they were treated by replacing 100 µL of supernatant with a complete medium containing individual compounds: polhxydroxysteroid glycosides PL1, PL2, and L1 at 12.5–50 µM and AaLs at 200–800 µg/mL for 24 h, or their combinations PL1, PL2 or L1 at 12.5 µL for 24 h and then AaLs at 10, 20, and 40 µg/mL for an additional 48 h. The 3D HCT 116 spheroid from each treatment was collected into the 1.5 mL microcentrifuge tubes and left for sedimentation for 10 min. Then, the supernatants were aspirated, 100 µL of cooled ECM gel from an Engelbreth-Holm-Swarm murine sarcoma (“Sigma-Aldrich”, Saint Louis, MO, USA) was added to the spheroid and transferred to 96-well plate covered by 100 µL of ECM gel (4 °C). The spheroids were maintained at 37 °C, in a 5% CO_2_ incubator for 96 h. A photo of the 3D HCT 116 spheroids (40 × 500 μm scale) was made at the time points of 0, 24, 48, and 72 h with the aid of a microscope Motic AE 20. A quantitative assessment of the invasion of spheroids was carried out using the ImageJ program and was determined as the difference between the total area invaded by the cells leaving the spheroid and the area of the spheroid as described previously [39].

#### 3.4.6. Western Blotting Assay

To determine the effect of the combined treatment with the investigated compounds on the activity of kinases and apoptotic proteins, the 3D HCT 116 spheroids were treated by combinations of PL1, PL2 or L1 at 12.5 µL and AaLs at 10, 20, and 40 µg/mL as described above in Section 3.4.5. “Determination of anti-invasive activity by 3D spheroids invasion assay”. The spheroids (*n* = 9) from each treatment were collected into the 1.5 mL microcentrifuge tubes and left for sedimentation for 10 min. The spheroids were washed by PBS and then lysed by 1X lysis buffer (“Cell Signaling Technology”, Danvers, MA, USA) according to the manufacturer’s protocol. The Lysate protein was extracted and blotted (20–40 µg) as described previously by Vishchuk et al. [40]. The membranes were treated with the primary antibodies against p-AKT (Ser 473), AKT, Bax, Bcl-XL, caspase 3, cleaved caspase 3, and β-actin (“Cell Signaling Technology”, Danvers, MA, USA) and a secondary antibody from a rabbit or a mouse (“Sigma-Aldrich”, St. Louis, MO, USA), according to the manufacturer’s protocol. Protein bands were visualized using an enhanced chemiluminescence reagent (ECL) (“Bio Rad”, Hercules, CA, USA) according to the manufacturer’s protocol.

#### 3.4.7. Combination Index (CI) Calculation

The polyhydroxysteroid glycosides/sulfated laminaran dose-effects were calculated by “Compusyn software” (ComboSyn, Inc.,) using the median effect method described by Chou [41].

#### 3.4.8. Statistical Analysis

All of the assays were performed in at least three independent experiments. Results are expressed as the mean ±standard deviation (SD). The Student’s *t* test was used to evaluate the data with the following significance levels: * *p* < 0.05, ** *p* < 0.01, *** *p* < 0.001.

## Figures and Tables

**Figure 1 marinedrugs-19-00540-f001:**
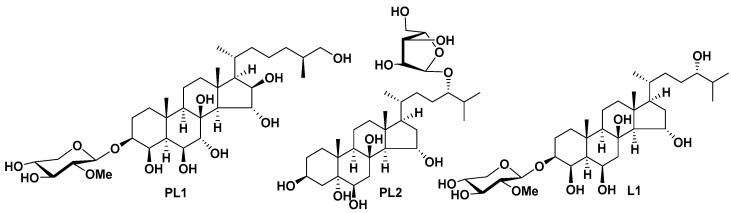
The structures of polyhydroxysteroid glycosides PL1, PL2, and L1 from *P. lincki*.

**Figure 2 marinedrugs-19-00540-f002:**
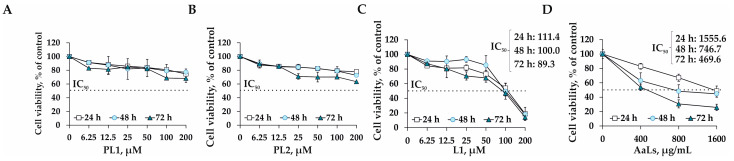
Individual effect of PL1, PL2, L1, and AaLs on viability and proliferation of 3D HCT 116 spheroids. The 3D HCT 116 spheroids were treated with (**A**) PL1, (**B**) PL2, (**C**) L1 (6.25–200 μM), and (**D**) AaLs (400–1600 μg/mL) for 24 h, 48 h, and 72 h. Cell viability was assessed using the MTS assay. All analyses were performed in three independent experiments. Results are expressed as mean ± standard deviation (SD).

**Figure 3 marinedrugs-19-00540-f003:**
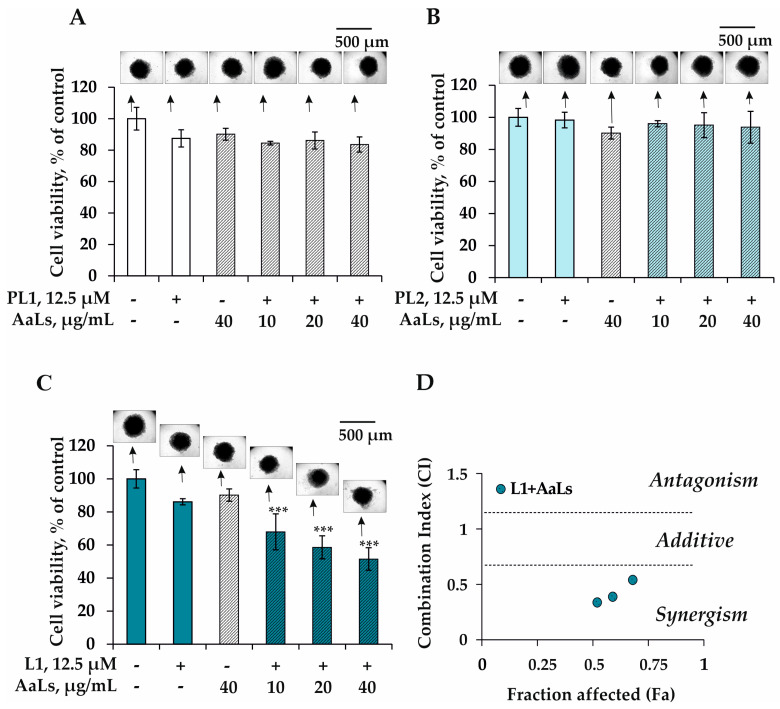
Combined effect of PL1, PL2, L1, with AaLs on viability and proliferation of 3D HCT 116 spheroids. The 3D HCT 116 spheroids were treated with PL1 (**A**), PL2 (**B**), and L1 (**C**) (12.5 μM) in combination with AaLs (10, 20, and 40 μg/mL) for 72 h. Cell viability was assessed using the MTS assay. Photos (*n* = 6 for control or cells treated by PL1, PL2, or L1 with AaLs, where *n* = the number of photos) of each spheroid were made using a ZOE™ Fluorescent Cell Imager and the analysis of spheroids was conducted using ImageJ software bundled with 64-bit Java 1.8.0_112 (NIH, Bethesda, Maryland, USA)). (**D**) Type of combination of L1 with AaLs calculated by Compusyn software 1.0.1 (ComboSyn, Inc., Paramus, NJ, USA). Combination index (CI) is a quantitative measure of the degree of interaction between different treatments. A CI equal to 0.9–1.1 is considered additive; a CI value of greater than 1.1 represents antagonism; and CI values less than 0.7 denotes synergism. All analyses were performed in three independent experiments. Results are expressed as mean ± standard deviation (SD). The Student’s *t* test was used to evaluate data with the following significance level *** *p* < 0.001.

**Figure 4 marinedrugs-19-00540-f004:**
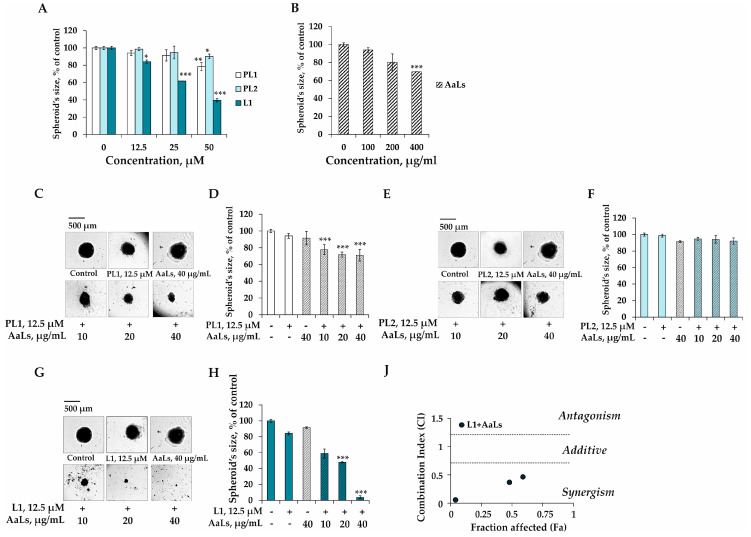
Individual and combined effects of polyhydroxysteroid glycosides from *P. lincki* and sulfated derivative of laminaran from *A. angusta* on the colony growth of 3D HCT 116 spheroids. The 3D HCT 116 spheroids were treated with (**A**) PL1, PL2, L1 (12.5, 25, 50 μM); (**B**) AaLs (100, 200, 400 μg/mL) or (**C**,**D**) PL1 (12.5 μM) in combination with AaLs (10, 20, 40 μg/mL); (**E**,**F**) PL2 (12.5 μM) in combination with AaLs (10, 20, 40 μg/mL); and (**G**,**H**) L1 (12.5 μM) in combination with AaLs (10, 20, 40 μg/mL) in soft agar. Photos of colonies were made using a Motic microscope (40× magnification, 500 μm scale). Results are expressed as mean ± standard deviation (SD). The Student’s *t* test was used to evaluate data with the following levels of significance: * *p* < 0.05, ** *p* < 0.01, *** *p* < 0.001. All analyses were performed in three independent experiments. (**J**) The Combination index (CI) calculated using the Compusyn software 1.0.1 (ComboSyn, Inc., Paramus, NJ, USA), is a quantitative measure of the interaction between L1 and AaLs in a 3D cell culture model. CI of less than 0.7 indicates a synergism of the effect of L1 and AaLs.

**Figure 5 marinedrugs-19-00540-f005:**
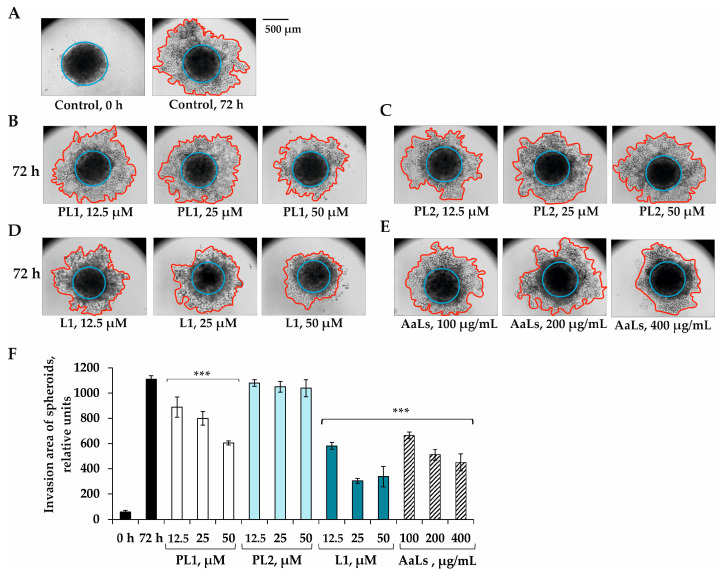
Individual effects of polyhydroxysteroid glycosides from *P. lincki* and sulfated derivative of laminaran from *A. angusta* on the invasion of 3D HCT 116 spheroids. The 3D HCT 116 spheroids were treated with (**A**) PBS (control) (**B**) PL1, (**C**) PL2, (**D**) L1 (12.5, 25, 50 μM) or (**E**) AaLs (200, 400, 800 μg/mL) after 72 h of invasion into the ECM gel. Photos of spheroids were made using a Motic microscope (40× magnification, 500 μm scale). (**F**) Quantitative measurement of the invasion area of spheroids treated with individual compounds was carried out using the ImageJ program and determined as the difference between total area invaded by cells leaving the spheroid and the area of the spheroid. All analyses were performed in three independent experiments. Results are expressed as mean ± standard deviation (SD). The Student’s *t* test was used to evaluate data with the following levels of significance *** *p* < 0.001.

**Figure 6 marinedrugs-19-00540-f006:**
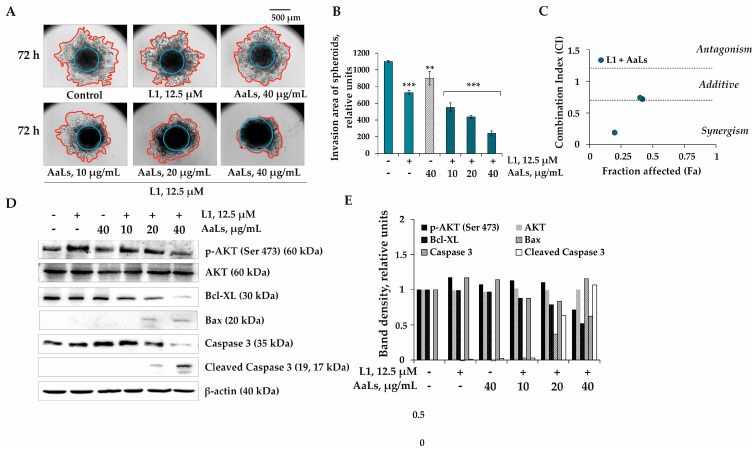
Combined effects of polyhydroxysteroid glycosides from *P. lincki* and sulfated derivative of laminaran from *A. angusta* on the invasion of 3D HCT 116 spheroids. (**A**) Photos (40× magnification, 500 μm scale) of 3D HCT 116 spheroids were treated with L1 (12.5 μM) in combination with AaLs (10, 20, 40 μg/mL) after 72 h invasion into the ECM gel. (**B**) Quantification of the invasion area of the spheroids was evaluted with the aid of ImageJ program. (**C**) The CI, calculated using the Compusyn software 1.0.1 (ComboSyn, Inc., Paramus, NJ, USA), is a quantitative measure of the interaction between L1 and AaLs in a 3D spheroids invasion model. (**D**) The regulation of the activity of AKT (Ser 473) kinase and apoptotic proteins Bcl-XL, Bax, caspase 3, and cleaved caspase 3 by L1 in combination with AaLs. (**E**) Relative band density was measured using Image Lab™ Software 4.1. Results are expressed as mean ± standard deviation (SD). The Student’s *t* test was used to evaluate data with the following levels of significance: ** *p* < 0.01, *** *p* < 0.001.

**Table 1 marinedrugs-19-00540-t001:** Structural characteristics of laminaran and its sulfated derivative from the brown alga *A. angusta*.

Polysaccharides from *A. angusta*	Yield, %	Mw, kDa	Content, % **	
			Carbohydrates	SO_3_Na^−^
AaL	0.8 *	5–6	98	0
AaLs	89 **	10–12	51	44

* % of defatted dried alga weight. ** % of sample weight taken for analysis.

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
