# Peer review of "Combined Anticancer Effect of Sulfated Laminaran from the Brown Alga Alaria angusta and Polyhydroxysteroid Glycosides from the Starfish Protoreaster lincki on 3D Colorectal Carcinoma HCT 116 Cell Line"

_marinedrugs, 2021, doi:10.3390/md19100540_

Round 1
Reviewer 1 Report
In the study “Combined anticancer effect of sulfated laminaran from the brown alga Alaria angusta and polyhydroxysteroid glycosides from the starfish Protoreaster lincki on 3D colorectal carcinoma cell HCT 116 cell line” Olesya S. Malyarenko et al investigated the anti-cancer activity of polysaccharide laminaran of brown algae and its sulfated derivative AaLs and the polyhydroxysteroid glycosides PL1, PL2 and L1 from the starfish P. lincki in colorectal carcinoma HCT 116 cell culture model. A significance of this study was that authors used 3D spheroid culture s of these cells to assess the antiproliferative activity of the compounds instead of 2D monolayer culture. The authors assessed the activities of these compounds individually and in combination to assess their effects on proliferation, colony formation and invasive activities of the HCT 116 cells. They also showed that the underlying mechanism of antiproliferative effects of the treatments involves suppression of pro-survival protein Bcl-XL protein and increased levels of pro-apoptotic protein Bax consequent to decreased activity of the serine/threonine kinase AKT kinase and activation of caspase-3.
Comments: Many natural products derived from plants and animals (terrestrial, fresh water or marine) have therapeutic biological activities against many pathological states. Their isolation, identification and biological activity screening has the potential to yield compounds with that are safe and highly specific. The study presented in the manuscript show that the polysaccharide laminaran from brown algae and hydroxysteroid glycosides from the starfish have anti-cancer activity against colorectal cancer cells. More significantly the authors show that sulfated laminaran compound AaLs show increased biological activity than the native compound and in combination the AaLs and steroid glycoside L1 from starfish show synergistic interaction in the biological activities assesses. This is a well-done study.
Specific comments:
- Although the authors showed the involvement of AKT kinase in the apoptotic action of these compounds, this may not the only biochemical and molecular effect of these compounds (particularly these compounds are known to exert a broad spectrum of therapeutic activities) that may be involved in their growth inhibitory action. There could be a number other signaling pathways they could have affected.
- Page 3 line 116 – nucleus not nuclus
- Page 5 line 181 – The authors….
Author Response
Dear Reviewer!
Thank you for careful review of our manuscript “Combined anticancer effect of sulfated laminaran from the brown alga Alaria angusta and polyhydroxysteroid glycosides from the starfish Protoreaster lincki on 3D colorectal carcinoma HCT 116 cell line”. We are very grateful for your censorious remarks and useful comments. The manuscript was revised and corrected according to the comments. Notably:
In the study “Combined anticancer effect of sulfated laminaran from the brown alga Alaria angusta and polyhydroxysteroid glycosides from the starfish Protoreaster lincki on 3D colorectal carcinoma cell HCT 116 cell line” Olesya S. Malyarenko et al investigated the anti-cancer activity of polysaccharide laminaran of brown algae and its sulfated derivative AaLs and the polyhydroxysteroid glycosides PL1, PL2 and L1 from the starfish P. lincki in colorectal carcinoma HCT 116 cell culture model. A significance of this study was that authors used 3D spheroid culture s of these cells to assess the antiproliferative activity of the compounds instead of 2D monolayer culture. The authors assessed the activities of these compounds individually and in combination to assess their effects on proliferation, colony formation and invasive activities of the HCT 116 cells. They also showed that the underlying mechanism of antiproliferative effects of the treatments involves suppression of pro-survival protein Bcl-XL protein and increased levels of pro-apoptotic protein Bax consequent to decreased activity of the serine/threonine kinase AKT kinase and activation of caspase-3.
Many natural products derived from plants and animals (terrestrial, fresh water or marine) have therapeutic biological activities against many pathological states. Their isolation, identification and biological activity screening has the potential to yield compounds with that are safe and highly specific. The study presented in the manuscript show that the polysaccharide laminaran from brown algae and hydroxysteroid glycosides from the starfish have anti-cancer activity against colorectal cancer cells. More significantly the authors show that sulfated laminaran compound AaLs show increased biological activity than the native compound and in combination the AaLs and steroid glycoside L1 from starfish show synergistic interaction in the biological activities assesses. This is a well-done study.
Comment 1. Although the authors showed the involvement of AKT kinase in the apoptotic action of these compounds, this may not the only biochemical and molecular effect of these compounds (particularly these compounds are known to exert a broad spectrum of therapeutic activities) that may be involved in their growth inhibitory action. There could be a number other signaling pathways they could have affected.
Answer 1. We agree with Reviewer’s comment that natural compounds including polysaccharides from brown algae and polyhydroxysteroid glycosides of the starfishes possessed various biological activities via modulating multiple signaling pathways [Ratovitski, E.A. Anticancer Natural Compounds: Molecular Mechanisms and Functions Part II. Current genomics. 2017, 18(2), 105. doi: 10.2174/138920291802170130191405]. In the present study the investigation of molecular mechanism of anticancer effect of individual compounds or their combination was focused on the regulation of AKT kinase activity and induction of apoptosis in colorectal cancer cells HCT 116, because AKT kinase is overexpressed and activated in colorectal cancer that lead to survival of cancer cells avoiding the apoptosis, and, consequently metastasis. We showed that one of the possible mechanisms of anticancer effect of sulfated laminaran AaLs and linkoside L1 is inhibition of phosphorylation of AKT, regulation of anti-apoptotic Bcl-XL and pro-apoptotic protein Bax expression level and activation executor of apoptosis caspase 3 resulting in apoptosis realization. This explanation is included in the text of the manuscript (Page 11) as “One of the possible ways of the human colorectal carcinoma progression and, consequently, its metastasis is the overexpression and activation of the survival factor serine/threonine kinase, AKT…”.
Comment 2. Page 3 line 116 – nucleus not nuclus
Answer 2. We agree with Reviewer’s comment. Corrected as suggested by Reviewer.
Comment 3. Page 5 line 181 – The authors….
Answer 3. We agree with Reviewer’s comment. Corrected as suggested by Reviewer.
We hope that you will be pleased with this revision. We wish to thank you for your patience and wisdom throughout this process.
Reviewer 2 Report
The manuscript entitled: “Combined anticancer effect of sulfated laminaran from the brown alga Alaria angusta and polyhydroxysteroid glycosides from the starfish Protoreaster lincki on 3D colorectal carcinoma HCT 116 cell line” presents an interesting study to the scientific community and proper consist in the scope of the journal.
Author Response
Dear Reviewer!
Thank you for careful review of our manuscript “Combined anticancer effect of sulfated laminaran from the brown alga Alaria angusta and polyhydroxysteroid glycosides from the starfish Protoreaster lincki on 3D colorectal carcinoma HCT 116 cell line”. We are grateful for the appreciation of our work.
Reviewer 3 Report
In this manuscript the authors evaluate anticancer effects of sulfated laminaran from the brown alga Alaria angusta and polyxydroxysteroid glycosides from the starfish Protoreaster lincki in 3D spheroids of colorectal carcinoma HCT 116 cells.
The topic is interesting, but there are some criticisms:
- the authors report in Figure 2 IC50 of L1 and AaLs and their value is very high. But much lower concentrations than IC50 were used in subsequent experiments. Why were the concentrations of 12.5 μM for L1 and 10, 20, and 40 μg/mL for AaLs chosen? Please explain
- in the different experiments it is evident that combined effect of L1 with AaLs is very effective. But I think more experiments would be needed and the data reported in the manuscript are preliminary. For example, use a PI3K inhibitor to confirm PI3K involvement in the action of L1 and AaLs , or demonstrate apoptosis not only by western blotting but also by other methods. A cell cycle analysis would be interesting
Author Response
Dear Reviewer!
Thank you for careful review of our manuscript “Combined anticancer effect of sulfated laminaran from the brown alga Alaria angusta and polyhydroxysteroid glycosides from the starfish Protoreaster lincki on 3D colorectal carcinoma HCT 116 cell line”. We are very grateful for your censorious remarks and useful comments. The manuscript was revised and corrected according to the comments. Notably:
In this manuscript the authors evaluate anticancer effects of sulfated laminaran from the brown alga Alaria angusta and polyxydroxysteroid glycosides from the starfish Protoreaster lincki in 3D spheroids of colorectal carcinoma HCT 116 cells.
The topic is interesting, but there are some criticisms:
Comment 1. The authors report in Figure 2 IC50 of L1 and AaLs and their value is very high. But much lower concentrations than IC50 were used in subsequent experiments. Why were the concentrations of 12.5 μM for L1 and 10, 20, and 40 μg/mL for AaLs chosen? Please explain
Answer 1. To found out the ability of sulfated laminaran AaLs to potentiate the anticancer activity of polyhydroxysteroid glycoside L1 we chose low non-toxic concentrations of individual compounds, at which they did not or slightly affect the viability of spheroids (this explanation is in the text of the manuscript (Page 5, 2nd paragraph)). The concentrations of individual compounds for combined treatment were chosen based on experimental data of cell viability assay. We preliminary checked the combined effect of AaLs at the concentrations of 10, 20, 40, 80, 160, 320 µg/mL and L1 at 6.25, 12.5, 25, and 50 µM on viability of 3D HCT 116 spheroids by MTS assay and concluded that optimal concentrations of AaLs is 10, 20, and 40 µg/mL and L1 is 12.5 µM for synergistic anticancer effect (data not shown in the manuscript).
Comment 2. In the different experiments it is evident that combined effect of L1 with AaLs is very effective. But I think more experiments would be needed and the data reported in the manuscript are preliminary. For example, use a PI3K inhibitor to confirm PI3K involvement in the action of L1 and AaLs , or demonstrate apoptosis not only by western blotting but also by other methods. A cell cycle analysis would be interesting.
Answer 2. Dear Reviewer, thank you very much for valuable remark. In this study we found out strong synergism of combined effect of sulfated laminaran AaLs and linkoside L1 in combating of colorectal cancer cells and bring to light the one of the possible mechanism of their action. The deep elucidation of the molecular mechanism of combined effect of investigated compounds as well as their effect in vivo is planned to be realized in our further work.
We hope that you will be pleased with this revision. We wish to thank you for your patience and wisdom throughout this process.
Round 2
Reviewer 3 Report
The authors did not respond to my review. Nothing has been explained or changed based on my comments.
My previous comments are:
In this manuscript the authors evaluate anticancer effects of sulfated laminaran from the brown alga Alaria angusta and polyxydroxysteroid glycosides from the starfish Protoreaster lincki in 3D spheroids of colorectal carcinoma HCT 116 cells.
The topic is interesting, but there are some criticisms:
- the authors report in Figure 2 IC50 of L1 and AaLs and their value is very high. But much lower concentrations than IC50 were used in subsequent experiments. Why were the concentrations of 12.5 μM for L1 and 10, 20, and 40 μg/mL for AaLs chosen? Please explain
- in the different experiments it is evident that combined effect of L1 with AaLs is very effective. But I think more experiments would be needed and the data reported in the manuscript are preliminary. For example, use a PI3K inhibitor to confirm PI3K involvement in the action of L1 and AaLs , or demonstrate apoptosis not only by western blotting but also by other methods. A cell cycle analysis would be interesting
Author Response
Dear Reviewer!
Thank you for careful review of our manuscript “Combined anticancer effect of sulfated laminaran from the brown alga Alaria angusta and polyhydroxysteroid glycosides from the starfish Protoreaster lincki on 3D colorectal carcinoma HCT 116 cell line”. We are very grateful for your remarks and useful comments which allow us improving our manuscript.
Comments and Suggestions:
The authors did not respond to my review. Nothing has been explained or changed based on my comments.
My previous comments are:
Comment 1. The authors report in Figure 2 IC50 of L1 and AaLs and their value is very high. But much lower concentrations than IC50 were used in subsequent experiments. Why were the concentrations of 12.5 μM for L1 and 10, 20, and 40 μg/mL for AaLs chosen? Please explain
Answer 1. The data on the cytotoxic and anticancer activities of linkoside L1 from the starfish Protoreaster linckii are very limited. Linkoside L1 at 100 µM was reported to inhibit the viability of RAW 264.7 murine macrophage cells by 50% after 24 h of cells’ incubation as determined by MTT assay [Malyarenko, T.V.; Kicha, A.A.; Kalinovsky, A.I.; Ivanchina, N.V.; Popov, R.S.; Pislyagin, E.A.; Menchinskaya E.A.; Padmakumarb, K.P.; Stonik, V.A. Four new steroidal glycosides, protolinckiosides A - D, from the starfish Protoreaster lincki. Chem. Biodivers. 2016, 13(8), 998–1007. doi: 10.1002/cbdv.201500336.]. In our study the concentrations of linkoside L1 and sulfated laminaran AaLs for combined treatment were chosen based on experimental data of MTS, soft agar, and 3D spheroids invasion assays. It was determined that the concentration of L1 at which it exerts a half of its maximal inhibitory effect on cell viability (IC50) is 89.3 µM after 72 h of treatment (MTS assay); the concentration of L1 at which it reduces the size of colony of 3D HCT 116 spheroids by 50 % (IC50 for colonies growth) is 38.0. µM after 72 h of treatment (soft agar assay), and the concentration of L1 at which it reduced the invasion area by 50% (IC50 for invasion) is 16.5 µM after 72 h of treatment (3D spheroids invasion assay). So, to study the efficacy of anticancer effect of L1 in combination with AaLs we tested L1 in minimal inhibiting concentrations of 6.25 and 12.5 µM.
As for sulfated laminaran, IC50 is 469.6 µg/mL, IC50 for colonies growth is more than 400 µg/mL, IC50 for invasion – 342.8 µg/mL after 72 h of treatment. Based to the previous published data on the effect of combined effect of sulfated laminaran from Dictyota dichotoma with X-ray radiation, where the radiosensitizing activity of sulfated laminaran was observed at low concentrations of 10, 20, and 40 µg/mL [Malyarenko, O.S.; Usoltseva, R.V.; Zvyagintseva, T.N.; Ermakova. Laminaran from brown alga Dictyota dichotoma and its sulfated derivative as radioprotectors and radiosensitizers in melanoma therapy. Carbohydr. Polym. 2019, 206, 539-547. doi: 10.1016/j.carbpol.2018.11.008.], we checked the ability of AaLs to potentiate the activity of L1 at concentrations of 10, 20, 40, 80, and 160 µg/mL, which are much lower than its IC50.
We added the Supplementary Figure 2 providing the data on the preliminary experiments for optimization of concentrations of individual compound for combined treatment and explanation to the text of the manuscript (Page 5, Paragraph 2). As shown in Supplementary Figure 2B, a strong synergism of anticancer effect was observed after the treatment of 3D HCT 116 spheroids by L1 at 12.5 µM and sulfated laminaran AaLs at 10, 20, and 40 µg/mL. We did not find a significant difference between the efficacy of the combined action of L1 (12.5 μM) and AaLs at higher concentrations of 80 and 160 μg/mL compared to the action of AaLs at low concentrations of 10, 20, and 40 μg/mL. So, we investigated the combined effect of L1 at 12.5 µM and AaLs at 10, 20, and 40 µg/mL.
Comment 2. In the different experiments it is evident that combined effect of L1 with AaLs is very effective. But I think more experiments would be needed and the data reported in the manuscript are preliminary. For example, use a PI3K inhibitor to confirm PI3K involvement in the action of L1 and AaLs , or demonstrate apoptosis not only by western blotting but also by other methods. A cell cycle analysis would be interesting
Answer 2. In this study we determined significant inhibiting effect of sulfated laminaran AaLs in combination with linkoside L1 on proliferation, colony formation, and invasion on the model of 3D spheroids using set of methods of cell biology such as MTS, soft agar, and 3D spheroids invasion assays. We elucidated one of the possible molecular mechanisms of the combined anticancer effect of investigated compounds by Western Blot, which is an unambiguous technique of cell and molecular biology using for identification of specific proteins and the influence of the compounds their activity or expression level.
We agree with Reviewer’s comment that ideally it will be better to use additional approaches such as use of PI3K inhibitor or cell cycle analysis to confirm the involvement of PI3K/AKT pathway and the induction apoptosis, respectively, in the action of L1 and AaLs. However this work was mainly aimed to reveal the efficacy of the combination of marine natural compounds in combating of colorectal cancer cells that was done completely. The sentence “However, the further additional experiments such as the use of PI3K inhibitor, cell cycle analysis, and DNA comet assay are needed to confirm the involvement of the PI3K/AKT pathway and induction of apoptosis in the combined anticancer effect of sulfated laminaran from the brown alga A. angusta and polyhydroxysteroid glycosides from the starfish P. linckii.” was added to the text of the manuscript to indicate the limitation of the results on the molecular mechanism in this study (Page 10, Paragraph 1).
We hope that all comments have been addressed and you will be satisfied with revised version of the manuscript.